# Quantitative Structure-Property Relationship (QSPR) of Plant Phenolic Compounds in Rapeseed Oil and Comparison of Antioxidant Measurement Methods

**Melanie Platzer** [1,2,*], **Sandra Kiese** [2], **Tobias Asam** [3], **Franziska Schneider** [2], **Thorsten Tybussek** [2], **Thomas Herfellner** [2], **Ute Schweiggert-Weisz** [2,4] **and Peter Eisner** [1,2,5]

1   ZIEL-Institute for Food & Health, TUM School of Life Sciences Weihenstephan, Technical University of Munich, Weihenstephaner Berg 1, 85354 Freising, Germany; peter.eisner@ivv.fraunhofer.de
2   Fraunhofer Institute for Process Engineering and Packaging IVV, Giggenhauser Str. 35, 85354 Freising, Germany; sandra.kiese@ivv.fraunhofer.de (S.K.); franziska.schneider@ivv.fraunhofer.de (F.S.); thorsten.tybussek@ivv.fraunhofer.de (T.T.); thomas.herfellner@ivv.fraunhofer.de (T.H.); ute.weisz@ivv.fraunhofer.de (U.S.-W.)
3   Carl Bechem GmbH, Weststr. 120, 58089 Hagen, Germany; asam@bechem.de
4   Institute for Nutritional and Food Sciences, University of Bonn, Meckenheimer Allee 166a, 53113 Bonn, Germany
5   Faculty of Technology and Engineering, Steinbeis-Hochschule, George-Bähr-Str. 8, 01069 Dresden, Germany
*   Correspondence: melanie.platzer@ivv.fraunhofer.de

**Abstract:** Natural antioxidants are known for their ability to scavenge free radicals and protect oils from oxidation. Our aim was to study the structural properties such as the number of hydroxyl groups and Bors criteria of phenolic substances leading to high antioxidant activity in oil in order to analyze common trends and differences in widespread in vitro antioxidant assays. Therefore, 20 different phenolic substances were incorporated into rapeseed oil and were measured using pressurized differential scanning calorimetry (P-DSC) and the Rancimat method. The Bors criteria had the highest influence on the antioxidant effect in rapeseed oil, which is why myricetin (MYR), fulfilling all Bors criteria, reached the highest result of the flavonoids. In the Rancimat test and P-DSC, MYR obtained an increase in oxidation induction time (OIT) of $231.1 \pm 44.6\%$ and $96.8 \pm 1.8\%$, respectively. Due to differences in the measurement parameters, the results of the Rancimat test and P-DSC were only partially in agreement. Furthermore, we compared the results to in vitro assays (ABTS, DPPH, FC and ORAC) in order to evaluate their applicability as alternative rapid methods. These analysis showed the highest correlation of the oil methods with the results of the DPPH assay, which is, therefore, most suitable to predict the antioxidant behavior of oil.

**Keywords:** oil stability measurements; antioxidant activity; flavonoids; phenolic acids; structure-activity relationship

## 1. Introduction

Autoxidation of oils is the decisive factor for their shelf life and results in the formation of free fatty radicals, hydroperoxides and peroxides, which lead to a decreased oil quality and the formation of off-flavors and off-odors [1–5]. Accordingly, the oxidation of oils plays an important role in the food industry, but it also affects the properties of technical oils, such as lubricants [6,7]. Autoxidation can be induced by UV radiation or singlet oxygen and occurs in four steps: chain initiation, chain propagation, chain branching and chain termination [8–10]. It is influenced by the oil's composition and storage conditions, such as free fatty acid content, light penetration and temperature, and can be slowed down or even prevented by the addition of antioxidants [1,11,12].

A common method to increase the oxidation stability of oils is the use of antioxidants. In addition to synthetic antioxidants such as butylhydroxytoluol (BHT) and butylhydroxyanisol (BHA), there are natural substances that are known for their radical scavenging

ability [7,13–21]. These can be classified into different subgroups based on their structure, which range from the low molecular weight substances such as phenolic acids to complex components such as flavonoids [22,23]. They are found in numerous fruits and vegetables and are usually obtained by extraction processes using organic solvents [1,24–27]. The antioxidant behavior of several plant extracts and commercially available natural antioxidants, as well as their comparison to synthetic antioxidants, has been studied extensively. Research has focused on the substitution of synthetic substances and investigated their efficacy by comparing different extracts [28–46].

There are several methods to measure the oxidation stability of oils, such as sensory analysis, spectroscopic or fluorescence methods, determination of acid and peroxide values and chromatographic methods [10]. Due to its simplicity, small sample size and automated evaluation, the Rancimat method is particularly popular and is used in various fields from the food industry to biodiesel production [4,47,48]. For the measurement, a sample is heated to a defined temperature. The volatile degradation products formed during oxidation are transferred by an air stream into a measuring cell filled with demineralized water. The device detects the increase in conductivity as a function of time, and the end of the measurement is initiated by the rapid increase in conductivity [4,47,49,50]. This test can be performed at a wide range of temperatures and is therefore suitable for a variety of samples [4,47,49,50].

Another popular method to determine oil stability is measurement by pressurized differential scanning calorimetry (P-DSC), where thermally induced transitions, such as decomposition, melting or the crystallization of samples, which lead to a change in heat capacity, are determined. Due to the increased pressure, the measurement time can be shortened significantly [51–58]. This makes the method suitable for samples that have been mixed with strong antioxidants [58,59]. However, the method is not frequently used because a calorimeter is not standard laboratory equipment [60,61].

In addition, the antioxidant behaviour of natural compounds is often analyzed using in vitro spectrophotometric assays. These are based on different reaction mechanisms and are influenced by various factors such as temperature, measurement duration, and pH. Accordingly, overestimation or underestimation of antioxidant activity may occur [62–64]. Since the extracts are mixtures of different antioxidant substances, the structure-activity relationship (SAR) of the individual substances is not addressed. The in vitro assays are particularly suitable for comparing samples of the same raw material obtained with different extraction parameters and can thus be very useful to evaluate different production steps of food processing [64,65]. However, there is no recognized standard method regarding the measurement and evaluation of these assays, which makes it difficult or even impossible to compare the measurement results with data from the literature [64,66]. Accordingly, these assays are repeatedly strongly criticized, and their use is discouraged [64].

Nevertheless, these assays continue to be used in various fields and, therefore, are still relevant for research studies [64,67–70]. They are popular methods in research and development as well as in the food industry for predicting the antioxidant activity of plant extracts or natural phenolic compounds because they are simple and quick to perform, require only small sample amounts, and simple laboratory equipment is sufficient [62–64,67–75].

Although the in vitro assays are commonly used in literature, and much is known about the reaction mechanisms and the influence of SAR, the correlation of their results with different types of application media, e.g., oils, and which assay should be preferred is still not clarified. Therefore, we determined the antioxidant activity of 20 different phenolic compounds (shown in Table 1) in a lipophilic matrix. After incorporating the antioxidants into rapeseed oil, we analyzed its oxidation induction time (OIT) using the Rancimat method and carried out P-DSC. By comparison of the two methods, the influence of the structural properties of the phenolic compounds on the oxidative stability was studied in more detail. Furthermore, these findings were compared to the SAR of in vitro assays (the 2,2′-azino-bis (3-ethylbenzothiazoline-6-sulfonic acid) (ABTS), 2,2-diphenyl-1-picrylhydrazyl (DPPH), Folin–Ciocalteu (FC) and oxygen radical absorbance capacity

(ORAC) assays) published previously [76–78] to experimentally identify a rapid test method suitable to predict the antioxidant activity of phenolic compounds in oil.

**Table 1.** Phenolic compounds analyzed in this study (phenolic acids and subgroups of flavonoids) along with reference standard names, sample codes and substituents (adapted with permission from Table 1 in [76–78]).

| Subgroups | Reference Standards | Code | Position and Substituents | | | | | | |
|---|---|---|---|---|---|---|---|---|---|
| **Phenolic acids** | | | **1** | **3** | **4** | **5** | | | |
| | Caffeic acid | CAA | $(CH_2)_2COOH$ | OH | OH | H | | | |
| | 3,4-dihydroxybenzoic acid | DBA | COOH | OH | OH | H | | | |
| | Ferulic acid | FEA | $(CH_2)_2COOH$ | $OCH_3$ | OH | H | | | |
| | Gallic acid | GAA | COOH | OH | OH | OH | | | |
| | 4-hydroxybencoic acid | HBA | COOH | H | OH | H | | | |
| | *p*-coumaric acid | PCA | $(CH_2)_2COOH$ | H | OH | H | | | |
| | Sinapic acid | SIA | $(CH_2)_2COOH$ | $OCH_3$ | OH | $OCH_3$ | | | |
| | Syringic acid | SRA | COOH | $OCH_3$ | OH | $OCH_3$ | | | |
| **Flavonols** | | | **2′** | **3′** | **4′** | **5′** | **3** | **5** | **7** |
| | Kaempferol | KAE | H | H | OH | H | OH | OH | OH |
| | Morin | MOR | OH | H | OH | H | OH | OH | OH |
| | Myricetin | MYR | H | OH | OH | OH | OH | OH | OH |
| | Quercetin | QUR | H | OH | OH | H | OH | OH | OH |
| **Flavanones** | | | **3′** | **4′** | **3** | **5** | **7** | | |
| | Hesperetin | HES | OH | $OCH_3$ | H | OH | OH | | |
| | Naringin | NAG | H | OH | H | OH | Rham, Glc | | |
| | Naringenin | NAN | H | OH | H | OH | OH | | |
| | Taxifolin | TAF | OH | OH | OH | OH | OH | | |
| **Dihydrochalcones** | | | **4** | **2′** | **4′** | **6′** | | | |
| | Phloridzin | PHD | OH | OH | OH | Glc | | | |
| | Phloretin | PHT | OH | OH | OH | OH | | | |
| **Flavanols** | | | **3′** | **4′** | **3** | **4** | **5** | **7** | |
| | (+)-catechin | CAT | OH | OH | OH | H | OH | OH | |
| | (−)-epicatechin | EPC | OH | OH | OH | H | OH | OH | |

## 2. Materials and Methods

The following chemicals and reference standards were obtained from Sigma-Aldrich (Steinheim, Germany): caffeic acid (CAA), (+)-catechin (CAT), 3,4-dihydroxybenzoic acid (DBA), (−)-epicatechin (EPC), ferulic acid (FEA), gallic acid (GAA), 4-hydroxybenzoic acid (HBA), hesperetin (HES), kaempferol (KAE), morin (MOR), myricetin (MYR), naringenin (NAN), *p*-coumaric acid (PCA), phloridzin (PHD), phloretin (PHT), quercetin (QUR), sinapic acid (SIA), siringic acid (SRA), taxifolin (TAF), Tween 85 and Span 85. The standard reference naringin (NAG) was obtained from Carl Roth (Karlsruhe, Germany), and copper powder was obtained from Merck (Darmstadt, Germany). Food-grade rapeseed oil from Brökelmann & Co. Oelmühle GmbH & Co. KG (Hamm, Germany) was used. For the experiments, all phenolic substances were dissolved in ethanol absolute.

These solutions were measured in the ABTS, DPPH, FC and ORAC assays. The results were published before and are reproduced here for comparison [76–78]. The measurements were performed according to the literature with some modifications, and detailed instructions can be found elsewhere [76–82].

For incorporation of the phenolic substance ethanol solutions, rapeseed oil was mixed with 0.15 % (*w/w*) Tween 85 and Span 85 and homogenized for 5 min at 18,000 min$^{-1}$ using

a T 25 easy clean control Ultra Turrax disperser from IKA-Werke GmbH & Co. KG (Staufen, Germany). Subsequently, the oil was heated to 50 °C and blended by adding 1510 µL of the phenolic solutions to obtain oils with seven concentrations of polyphenols (0.075, 0.1, 0.25, 0.35, 0.5, 0.75 and 1 mM). The mixtures were first stirred for 2 min and then dispersed for another 2 min using the Ultra Turrax at 16,000 min$^{-1}$.

For the determination of OIT, 3 g of each oil sample were measured in duplicate together with 300 mg copper powder in a fully automated Rancimat 743 from Metrohm AG (Herisau, Switzerland). A temperature of 60 °C and a gas flow of 20 Lh$^{-1}$ were used. The reaction curves were generated automatically by Rancimat Metrodata version 1.1 software and analyzed using the tangent method [4,47,49,50]. In addition, the oil samples were measured at two concentrations (0.5 and 1 mM) in a P-DSC. These measurements were kindly provided by Carl Bechem GmbH using an ASTM standard (ASTM D6186-19) with some modifications. For the measurements, 1 mg of each sample was subjected to 7 bar and measured at 210 °C. For both measurements, the increase in oxidation stability was calculated compared to the blank (rapeseed oil blended with ethanol, Tween t85 and Span 85).

For both the Rancimat and P-DSC measurements, all prepared solutions of phenolic substances were measured, and the values were calculated for a 1 mM formulation. To compare the results of the OIT measurements and the antioxidant assays, the mean values and standard deviations of 1 mM solutions are shown in Figure 1.

For statistical analysis, one-way analysis of variance (ANOVA) was performed using Sigma Plot (Systat Software, San Jose, CA, USA) based on an unpaired *t*-test. In case of a significant difference, an additional paired test was performed using the Holm–Šidák method. The significance level for both tests was $p = 0.050$, and the statistical analysis was always carried out with all significant decimal places. The statistical analyses were performed for all measured substances. For discussions within phenolic subgroups, the statistical analysis within each group was also performed and referenced at the appropriate points in the text. Multivariate analysis was performed by principal component analysis using OriginPro2018.

## 3. Results and Discussion

### 3.1. Analysis of Oil Stability

The increase in OIT of the phenolic compounds in rapeseed oil (calculated for the concentration of 1 mM) using the racimat method and P-DSC are shown in Figure 1.

In both measurement methods, all substances led to an increase in OIT, with the exception of NAG, which showed no effect in the Rancimat measurements. Values in the range of 24.5 ± 4.3% to 231.1 ± 44.6% were obtained for the Rancimat test and 5.5 ± 2.0% to 145.9 ± 16.1% for the P-DSC measurements. Although the results of the two methods were determined by using the same evaluation procedure, different results were obtained, which can be explained by the measurement conditions used. Since the temperature in the P-DSC was much higher than in the Rancimat method, the values obtained were lower. It is known that temperature in particular can have a decisive impact on the stability of phenolic substances and can result in a decrease in antioxidative effect [83]. In some cases, a decrease in antioxidant activity was found even at temperatures of 40 and 60 °C, while in other studies, a decrease was found only at higher temperatures of 100 and 120 °C [83–88]. Thereby, the type of phenolic compounds determines the influence of the temperature. Flavonoids are considered to be particularly temperature-sensitive, and in particular, the subgroups of flavones, flavanols, and flavanones are known to be damaged even at lower temperatures [83,85,89–95]. Accordingly, the numerical values of the two measurement methods are not comparable, and therefore, only the sequences of individual substances will be discussed in the following.

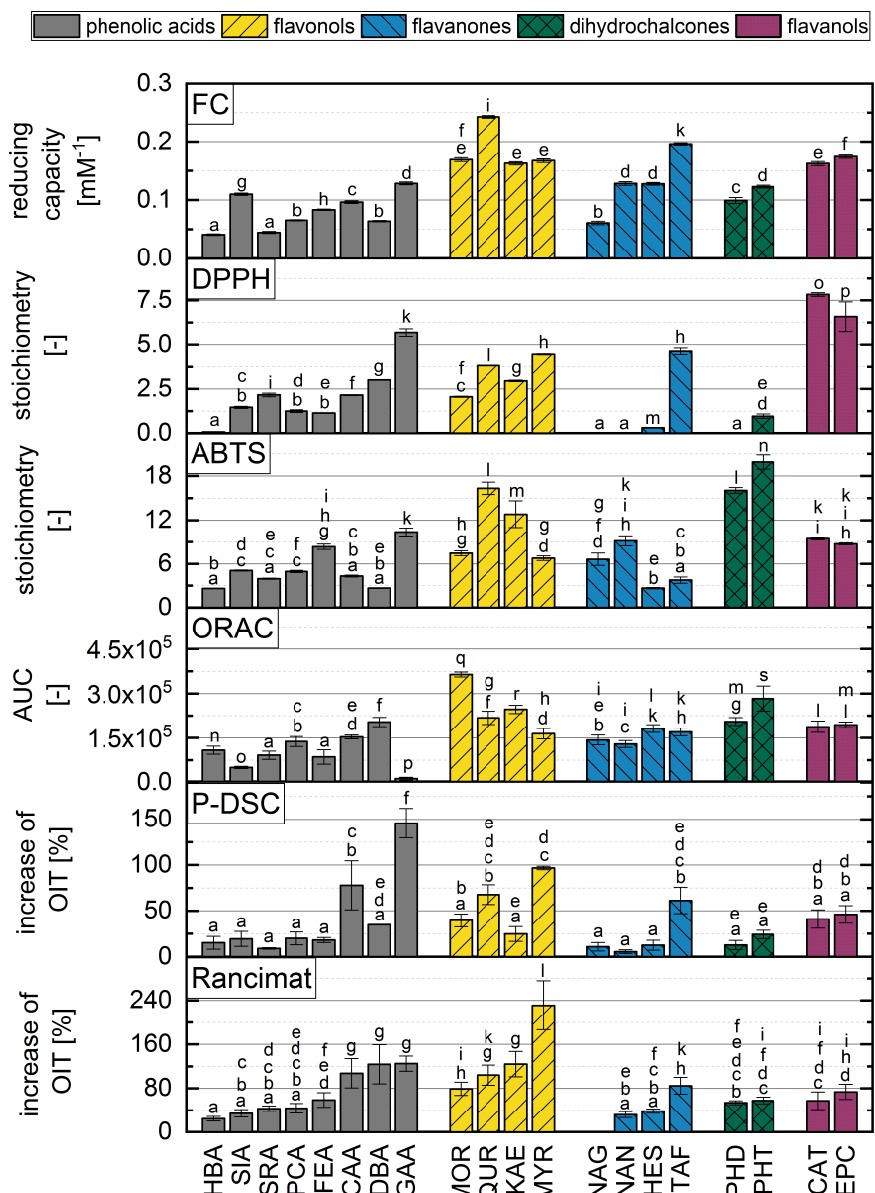

**Figure 1.** OIT increase in relation to rapeseed oil of all standard reference compounds determined by Rancimat method and P-DSC in comparison to ORAC, ABTS, DPPH and FC assay results (adapted with permission from Platzer et al. [76–78]). The results are shown as mean values with standard deviations for the concentration of 1 mM for all samples and all measured methods. Equal letters indicate that there is no significant difference between the results with an significance level at $p > 0.05$.

In the ranicmat measurements, the highest result was obtained by MYR, which fulfills all three Bors criteria. CAT and EPC, with only Bors 1, obtained lower values than MOR, KAE and TAF, each having two of the three criteria. Thus, for the Rancimat method, the more Bors criteria that were met, the higher the OIT. An exception is QUR, which fulfills all three Bors criteria, but showed a significantly reduced increase in OIT compared to MYR. QUR was more similar to substances with a catechol (CAA, DBA) or galloyl group (GAA), corresponding to Bors 1. Other phenolic acids containing no structural similarity to Bors criteria showed very low values.

In the P-DSC measurements, MYR and QUR showed higher results than KAE, EPC, CAT and MOR. Accordingly, substances fulfilling all Bors criteria seem to be particularly effective. CAT and EPC achieved the same result as KAE and MOR, which suggests that it does not matter whether a substance fulfills only Bors 1 or Bors 2 and 3. The highest result

in the P-DSC was obtained by GAA. Phenolic acids with a catechol group also obtained high results (CAA and DBA). It can be concluded that structural properties with high numbers of hydroxyl substituents, such as catechol or galloyl groups, have an influence on the effect in the oil not only for flavanoids.

### 3.2. Consolidated Analysis of Phenolic Subgroups

In the following section, the sequences in the individual subgroups are discussed, and additionally a significance test is performed within each subgroup.

Considering the phenolic acids, the highest increase in OIT in the Rancimat method was obtained for CAA, DBA and GAA. Since they were not significantly different, it does not seem to make a difference whether a substance has a catechol or a galloyl group. Due to the additional hydroxy group, a higher result for GAA would be expected. However, the opposite effect was already reported in some studies, where the presence of a galloyl group even led to a decrease in antioxidant behavior [96–99]. It is also known from the literature that galloyl groups are used for the formation of superoxide anions with the formation of stable radicals. Accordingly, compounds with this group often have a prooxidant effect, which could explain the outcomes of our study [100]. However, this could also be explained by steric hindrance due to the three substituents on the aromatic ring, as shown previously [76,78]. Moreover, compounds with a catechol group reached a higher increase in OIT than compounds with only one hydroxyl group (DBA and CAA higher than HBA, SIA, SRA, PCA, and FEA). The presence of these groups results in increased stability of the aromatic ring due to electron delocalization leading to high antioxidant activity [101–105]. Furthermore, HBA and SRA did not achieve significantly higher values than PCA and SIA, and it can be concluded that the prolongation of the electron delocalization system also has no influence on the antioxidant effect in oil. It was expected that the hydroxycinnamic acids would reach a higher oxidation stability than the hydroxybenzoic acids due to their additional conjugated double bond. This additional bond leads to an extended electron delocalization and a higher stability of the radical [106–109]. HBA and PCA achieved the same results as SRA, SIA and FEA, so an additional methoxy group also has no influence on the antioxidant effect in oil, which is not in agreement with the literature [98,99] Since these substances, just like GAA, also have three substituents, this result can possibly also be explained by steric hindrance. Methoxy groups are electron donors and thus reduce the dissociation energy, allowing electrons to be released more easily and increasing the antioxidant effect [98,110–112]. However, a lower antioxidant activity was found for FEA compared to CAA, which has a methoxy group instead of a hydroxyl group. This leads to a decrease in active electrons and hydrogen-donating groups and a lower antioxidant activity [106–109,113].

In the P-DSC measurements, the highest value was found for GAA. Due to the presence of a galloyl group, the molecule has the ability to release hydrogen atoms, which is particularly important for hydrogen transfer and is a reason for its high antioxidant activity [98,99]. Since this value was higher than the values of DBA and CAA, an additional hydroxyl group seems to positively influence the increase of OIT. Like the Rancimat results, DBA and CAA reached higher values than HBA and PCA, which only have one hydroxyl group. The more hydroxyl groups a molecule owns, the higher the antioxidant activity, which was also shown before [106,114]. HBA and SIA as well as SRA and PCA did not show significantly different values, which is why an additional methoxy group has also no influence on the antioxidant behavior in oil.

For the flavonols, the highest result in the Rancimat method was obtained for MYR, and therefore, a galloyl group increases the antioxidant effect in this measurement the most, which is in agreement with the literature [115]. A high increase in OIT was expected because the substance meets all of the Bors criteria and, therefore, should achieve the most efficient electron delocalization [116,117]. QUR and KAE did not show significantly different values, and the influence of the catechol group seems to be less strong. Here, a different result was expected, since the delocalization of the B-ring leads to an increased stability of the

phenoxyl radical. In addition, the delocalization of the B-ring is further enhanced by the formation of an intramolecular hydrogen bond by the catechol group [102,107,118–121]. The lowest antioxidant effect was obtained for MOR, which is why a negative influence by an additional hydroxyl group at C2′ cannot be excluded (KAE > MOR). This was not expected and is not consistent with the literature [122]. Similar results were obtained for the P-DSC measurements. Since MYR and QUR showed the highest values, the strongest influence seems to be provided by meeting Bors 1. MYR and MOR with 2 and 3 hydroxyl groups reached higher values than KAE with only one hydroxyl group, indicating that the number of hydroxyl groups on the B-ring has a decisive influence on the effect of the substances in oil. In addition to the number, the position of the hydroxyl groups also influences the antioxidant behavior [110,111,123,124]. QUR with one catechol group (corresponding to Bors 1) reached a higher value than MOR, which has one hydroxyl group at C2′ and C4′ of the B-ring.

TAF reached the highest value for the group of flavanones, which is the only substance with Bors 1. An influence of the hydroxyl group on C3 cannot be confirmed by these results, since this property is only fulfilled by TAF. It is therefore possible that, in addition to the catechol group on the B-ring, the C3-OH also has a positive influence on the antioxidant behavior in oil. The higher result of TAF compared to HES was expected, since it is already known from the literature that a hydroxyl group in the *para* position has a positive influence on the antioxidant activity [79,98,99,111,123,125–127]. NAN obtained a higher result than NAG, indicating that the presence of a hydroxyl group at C7 has a positive influence on the antioxidant effect. NAG was the only substance that showed no reaction in the Rancimat method. The reason for this could not be clarified based on our experiments, but it is possible that this effect can be attributed to the sugar residue. In the P-DSC measurements, only TAF reached a significantly higher result than the other substances, again due to Bors 1. NAN, NAG and HES induced the same increase in OIT; therefore, neither the presence of a hydroxyl group at C7 nor an additional methoxy group at C4′ seems to affect the antioxidant activity.

For the dihydrochalcones as well as for the flavanols, both OIT tests showed the same sequences of results. PHD and PHT led to significantly the same increase in OIT, so an additional OH group at C6′ does not affect the antioxidant behavior in the oil. Furthermore, it can be concluded from the results of the flavanols that the structural spatial arrangement also has no influence on an antioxidant effect in the oil. Accordingly, both tests showed the same values for CAT and EPC, which was in agreement with literature [128].

In summary, the results of the P-DSC and the Rancimat measurements show similar trends. Both high and low correlation between these two methods have been reported in the literature [129–135].

### 3.3. Comparision of Oil Measurement Methods with In Vitro Antioxidant Assays

In order to identify possible rapid test methods to predict the antioxidant effect in oils, the results of the OIT measurements are compared to various in vitro assays, which were previously published [76–78] and are discussed in the following. To simplify comparability, the previously published measurement results of the same phenolic substances from the ABTS, DPPH, FC, and ORAC assays are presented once again, and the results are reproduced from Platzer et al. [76–78] in Figure 1. The in vitro results of MYR were not reported before and were measured for this study using the same measurement and evaluation methods.

For the OIT measurements, a strong influence of the Bors criteria on the antioxidant behavior was found. Furthermore, phenolic substances fulfilling only Bors 1 were similar to substances with Bors 2 and 3. The same behavior was also observed in the FC and DPPH assays, whereas these results were not observed in the ABTS and ORAC assays. In the literature, correlations as well as differences of antioxidant assays with each other and in comparison to oil measurements are reported [62,64,76–78,126,136,137]. In general, it is known that the antioxidant behavior of a substance depends on the position and

the number of the hydroxyl groups. Furthermore, it is also known that the nature of the substituents of a molecule influences the antioxidant behavior more than its basic structure [107,109,111,138–142].

For the phenolic acids, the ABTS, FC and ORAC assays showed higher antioxidant activities for substances with a hydroxycinnamic acid group than for substances with a hydroxybenzoic acid group, which is not in agreement with the results of the DPPH assay and the oil measurements. As already known from the literature, the number of hydroxyl groups has an influence on the antioxidant behavior. This was confirmed in the DPPH and FC assays, as well as in the OIT measurements. The more hydroxyl groups a substance has, the higher the antioxidant effect. These results were not shown for the ORAC and the ABTS assays. Here, substances with one galloyl group sometimes even achieved lower values, which may be due to steric hindrance. The influence of an additional methoxy group also affects the methods differently. While this has no influence on the antioxidant behavior in the ABTS, DPPH, FC assays as well as in the OIT measurements, it even positively influenced the results obtained in the ORAC and ABTS assays.

Additionally, for the group of flavonols, different results were found. While the presence of Bors criteria does not play a role for the results in the ABTS and ORAC assay, the influence on the remaining assay seems to be even higher. Both the number and the type of Bors criteria is crucial here. In the DPPH and FC assays as well as in the OIT measurements, the highest influence was shown by Bors 1. Thus, substances with a catechol group on the B-ring achieved higher antioxidant activities than substances with only an hydroxyl group. In the DPPH assay and in the P-DSC measurements, substances with a galloyl group also showed the highest activity. The fact that this result cannot be transferred to the other test methods may be due to steric hindrance.

In all measurement methods, a positive influence of Bors 2 and 3 on the antioxidant effect was found for the flavanones. While a positive effect of a hydroxyl group in the para position was also found for all in vitro tests, this result could not be conclusively confirmed in the oil measurements, but a positive influence is probable. The presence of a sugar residue at C7 showed a slightly negative influence in all methods or did not affect the antioxidant activity. Furthermore, an additional methoxy group showed no effect on the antioxidative behavior in any method, with the exception of the ORAC assay. In the DPPH, FC and ORAC assays as well as in the two OIT measurements, the highest influence was observed for Bors 1.

In all in vitro methods, a higher antioxidative behavior was found for a substance with an additional hydroxyl group at C6'. This was not the case for the two OIT measurements. Here, no significantly different results were obtained, which means that a positive effect of the hydroxyl group, just like a negative one of the sugar residue, can be excluded. In all methods, a different three-dimensional arrangement of two structural isomers does not seem to influence the result. The two flavanols gave similar activities in all tests.

*3.4. Principal Component Analysis of the DPPH, ABTS, FC, and ORAC Assays and OIT Measurements*

To better identify the correlations and differences of the in vitro and OIT measurements, principal component analysis was performed. First, a principal component (PC) analysis was carried out on all measured substances, and the results were distinguished with respect to the number of hydroxyl groups (Figure 2) and their different structural properties together with the respective subgroup (Figure 3).

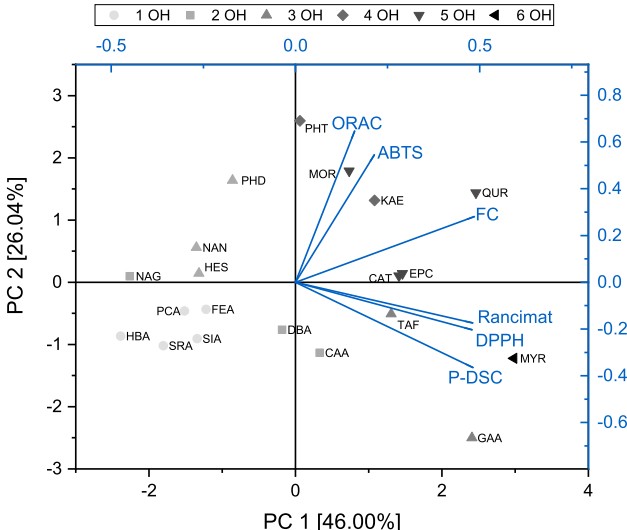

**Figure 2.** PC analysis of results from Rancimat and the P-DSC, ABTS, DPPH, FC, and ORAC assays presented in dependence of number of hydroxyl groups.

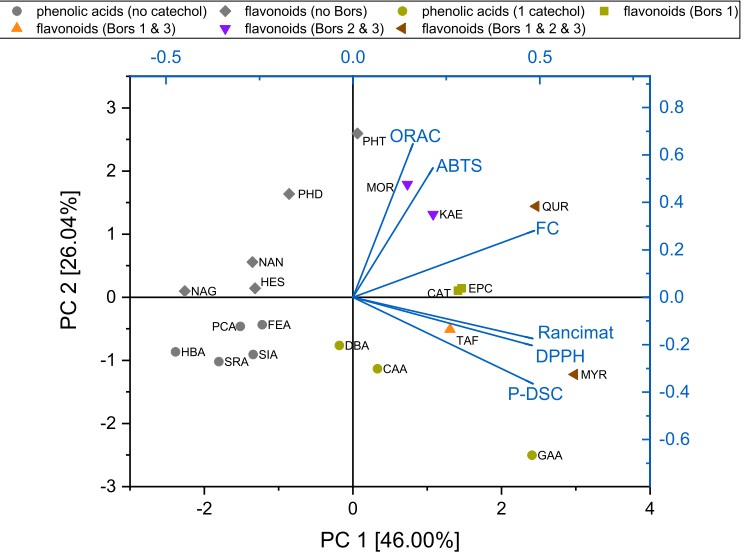

**Figure 3.** PC analysis of results from Rancimat and the P-DSC, ABTS, DPPH, FC, and ORAC assays presented in dependence of subgroups and structural properties.

Both PCs cover 72.04% of the initial variance of the original dataset. For PC1, both OIT and all in vitro measurements showed positive values. Therefore, PC1 corresponds to the total antioxidant effect, and a high score value in PC1 indicates a high activity. Both OIT measurements showed high positive loading values of 0.48 for PC1. This means that they strongly correlated to the total antioxidant effect. Additionally, the FC (0.49) and DPPH assays (0.48) showed high positive loading values for PC1. Therefore, these measurements are very suitable for the determination of the total antioxidant activity. PC2 showed a separation of the different methods. The ORAC, ABTS and FC assays showed positive loading values of 0.65, 0.55 and 0.28, respectively. In contrast, the Rancimat, DPPH and P-DSC measurements showed negative loading values of −0.17, −0.20 and −0.37, respectively. This indicates that the different methods may be based on different reaction mechanisms. Under this assumption, the ORAC, the ABTS and the FC assays possibly react according to a similar reaction mechanism. However, it should be noted that the loading plots of the ORAC and ABTS shows a higher correlation with each other than with the FC assay. Furthermore, the results show that the loading plots of the OIT measurements correlate well with that of the DPPH assay. Thus, the DPPH assay is the appropriate rapid

test to predict the effect of antioxidants in the oil. Such a correlation has also been reported in the literature for the measurement of nut oils [143].

Figure 2 also shows the influence of the hydroxyl groups on the antioxidant effect. Substances with a low number of hydroxyl groups showed lower values for PC1 and PC2 than substances with a high number, which means that the number of hydroxyl groups affects the values of all methods. Considering this observation, the ORAC, ABTS and FC assays showed a higher correlation to the number of hydroxyl groups than the other measurements. However, there are also some exceptions, such as GAA, which indicates that there are other influencing factors. These are shown in Figure 3, where the PC analysis is presented in dependence of subgroups and different structural properties. Here, it can be shown that substances without a catechol group and/or none of the Bors criteria also have the lowest antioxidant effect. Substances with a catechol group, on the other hand, have a higher antioxidant effect. It can also be concluded that substances fulfilling the first Bors criterion show a similar antioxidant effect than substances fulfilling Bors 2 and 3. Thus, Bors 1 seems to have the strongest influence on the antioxidant behavior for phenolic compounds. Since the results for substances fulfilling Bors 1 and 3 show a similar antioxidant effect than the substances fulfilling only Bors 1, the influence of Bors 3 criterion seems to be rather small. The highest PC1 value and thus also the strongest antioxidant effect was shown by substances fulfilling all three Bors criteria. Additionally, the influence of a galloyl group also has a strong effect on the antioxidant behavior in most methods, which is why GAA also achieved a high value, as well as MYR.

To discuss the structural differences and similarities between flavonoids and phenolic acids, PC analysis was repeated for these two groups (Figures 4 and 5). The FC and DPPH assays showed very similar loading values as the Rancimat and P-DSC measurement, which means that these four measurements lead to similar results for prediction of the antioxidant activity of flavanoids. Both the ABTS and the ORAC assay are neither suitable for measuring the antioxidant effect of flavonoids nor for predicting their effect in oil. Just as in the other PC analysis, substances that fulfill none of the Bors criteria show the lowest antioxidant activity, and substances that fulfill all three Bors criteria show the highest activity. In addition, Bors 1 seems to have the highest influence, since substances achieved similar values to substances fulfilling Bors 1 and 3 or Bors 2 and 3. It can also be concluded that the influence of Bors 3 is rather low.

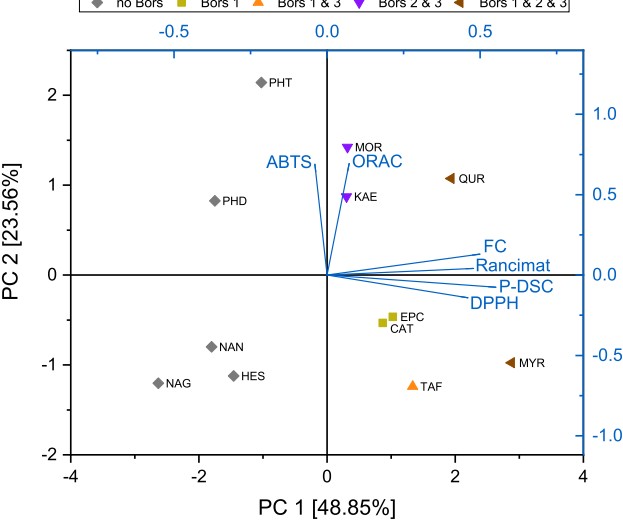

**Figure 4.** PC analysis of results of flavonoids from Rancimat and the P-DSC, ABTS, DPPH, FC, and ORAC assays presented in dependence of Bors criteria.

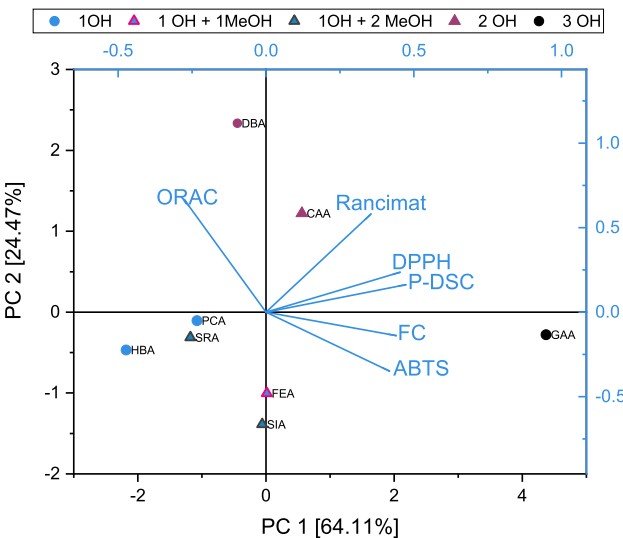

**Figure 5.** PC analysis of results of phenolic acids from Rancimat and the P-DSC, ABTS, DPPH, FC, and ORAC assays presented in dependence of hydroxyl and methoxy groups.

Figure 5 shows the results of the PC analysis for the phenolic acids. This analysis differs strongly from the analysis shown above. Both OIT measurements, Rancimat and P-DSC, showed similar but slightly different loading plots. Thus means, that for measurement of the antioxidant effect of polyphenols, both measurements will lead to different results. The DPPH assay shows loading values in between the Rancimat and P-DSC method. Therefore, the DPPH assay is the most suitable assay to predict the antioxidant effect of a phenolic acid in oil. Additionally, it can be concluded that the ORAC is less suitable to determine the antioxidant effect of phenolic acids. Considering the results of the individual substances, the values seem to depend on the number of substituents. The lowest results were obtained for substances with only one hydroxyl group and the highest for GAA, which has three hydroxyl groups. From the results, it is clear that additional methoxy groups also have a positive influence on the antioxidant behavior of the phenolic acids.

## 4. Conclusions

In our study, it was shown that especially the Bors criteria influence the antioxidant behavior of phenolic substances in oil. Basically, the more Bors criteria are fulfilled, the higher the antioxidant effect, with the first Bors criterion having the strongest influence. It was also shown that the results of the Rancimat method are partially in agreement with those from P-DSC. Differences in the two methods can be justified by different temperatures and other measurement parameters.

By performing different PC analyses, the antioxidant effects of phenolic substances in oil were compared with the antioxidant behavior in different in vitro antioxidant assays. There was an appropriate correlation of the OIT data with the results of the DPPH assay. Thus, this represents a good alternative to the oil methods and is suitable for predicting the antioxidant behavior in oil. This is a great advantage since the DPPH assay is a rapid method that can be performed in any conventional laboratory without complicated analytical equipment. In addition to cost and time savings, this also reduces the amount of sample required. Another advantage of the DPPH assay is that no complex sample preparation is required. Additionally, the FC showed a high correlation with the OIT values, so this assay could also be an alternative method. However, when PC analysis is performed only for the phenolic acid subgroup, different results are shown. Here, the OIT measurements in addition to the DPPH and FC assay also correlate with the antioxidant activity of the ABTS assay, but the correlation between the assays is not as clear as in the other PC analysis.

In summary, the in vitro antioxidant assays are suitable for predicting the antioxidant activity of polyphenolic standard references in oil. In our results, the highest correlations for the different phenolic subgroups were found for the DPPH assay. However, to use the method for the characterisation of natural extracts, further influences, e.g. pH, sugar and protein content, on the correlation have to be assessed first.

**Author Contributions:** Conceptualization, M.P. and S.K.; methodology, M.P., S.K., T.T. and T.A.; data curation, M.P., F.S. and T.A.; writing—original draft preparation, M.P.; writing—review and editing, S.K., T.A, F.S., T.T., T.H., U.S.-W. and P.E.; visualization, M.P. and S.K.; supervision, S.K., T.H., U.S.-W. and P.E.; funding acquisition, T.H. All authors have read and agreed to the published version of the manuscript.

**Funding:** This research was partly funded by the German Federal Ministry of Education and Research (grant numbers: 031B0387A and 031B0360A).

**Institutional Review Board Statement:** Not applicable.

**Informed Consent Statement:** Not applicable.

**Data Availability Statement:** Data of the measurement results are available from the authors.

**Conflicts of Interest:** The authors declare no conflict of interest. The funders had no role in the design of the study; in the collection, analyses, or interpretation of data; in the writing of the manuscript, or in the decision to publish the results.

## Abbreviations

| | |
|---|---|
| ABTS | 2,2'-azino-bis (3-ethylbenzothiazoline-6-sulfonic acid) |
| BHT | butylhydroxytoluol |
| BHA | butylhydroxyanisol |
| CAA | caffeic acid |
| CAT | (+)-catechin |
| DBA | 3,4-dihydroxybenzoic acid |
| DPPH | 2,2-diphenyl-1-picrylhydrazyl |
| EPC | (−)-epicatechin |
| FC | Folin–Ciocalteu |
| FEA | ferulic acid |
| GAA | gallic acid |
| HBA | 4-hydroxybenzoic acid |
| HES | hesperetin |
| KAE | kaempferol |
| MOR | morin |
| MYR | myricetin |
| NAG | naringin |
| NAN | naringenin |
| OIT | oxidation induction time |
| ORAC | oxygen radical absorbance capacity |
| P-DSC | pressurized differential scanning calorimetry |
| PC | principal component |
| PCA | *p*-coumaric acid |
| PHD | phloridzin |
| PHT | phloretin |
| QUR | quercetin |
| SIA | sinapic acid |
| SAR | structure-activity relationship |
| SRA | siringic acid |
| TAF | taxifolin |

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
