# Peer review of "Quantitative Structure-Property Relationship (QSPR) of Plant Phenolic Compounds in Rapeseed Oil and Comparison of Antioxidant Measurement Methods"

_processes, doi:10.3390/pr10071281_

Round 1

Reviewer 1 Report

Suggestions

Line 35: when we use the expression “has been studied extensively”, please, indicates these studies in reference list to improve this affirmation

Line 33-36: please, merge these two different sentences, because the second is meaningless like that and isolated in the text

In Material and Methods section, the protocol used to ABTS, DPPH, FC, ORAC assays can be described, including steps for oil extraction and analysis of each method applied

Line 111-112 and line 129-130: “, the oil samples were measured at two concentrations (0.5 and 1 mM) in a P-DSC” and “The increase in OIT of the phenolic compounds in rapeseed oil (calculated for the concentration 1 mM) using racimat method”, because the oils were analyzed only in these concentrations? Also, in the Figure 1 the data about the rapeseed oils and different antioxidant substances are represented in each concentration, this was not clear, because in Material and Methods section “phenolic solutions to obtain oils with seven concentrations of polyphenols (0.075, 0.1, 0.25, 0.35, 0.5, 0.75 and 1 mM)” and also, “at two concentrations (0.5 and 1 mM) in a P-DSC”. Please, the authors can provide this correct information. The data in Figure 1 is an average value? The question about phenolic concentrations in rapeseed oil can be clear to understand the results obtained.

Line 369-374: this paragraph can be changed to Material and Methods section or Results and Discussion section, not in Conclusion.

The manuscript showed a very good data, but a few information is needed in some points, also the Conclusion showed are supported by experiment data obtained in this study.

Reviewer 2 Report

The paper is well written and it will provide to readers the needed understanding of antioxidant activities in vegetable oils and alternative methods in measuring those activities.

Author Response

Dear Reviewer, thank you for your time and kind comments.

Reviewer 3 Report

The manuscript Quantitative structure-property relationship (QSPR) of plant phenolic compounds in rapeseed oil and comparison of antioxidant measurement methods is of current interest as it is a very comprehensive study.

In addition, several important aspects are taken into account when evaluating the structural properties contributing to antioxidant activity.

The introductory part is very comprehensive and the goal is well formulated, supported by the structure of the included information.

What is the main advantage of the manuscript, an important issue is the contribution of the structures and the discovery of the reasons for the developed activities, especially in the antioxidant activity as such.

I have the following question to the authors:

 - Why only 20 phenolic substances were chosen to be incorporated into rapeseed oil and those in particular?

- Regarding the analyzes performed, there are several additional titrimetric assays to determine lipid oxidation, but I believe that the Rancimat test is informative enough to be performed on its own. Therefore, I have no objections in this regard.
